# Inframammary Fold Banking of the Non-Dominant Superficial Epigastric Vein (SIEV) in Unilateral Autologous Breast Reconstruction: A Simple and Helpful Backup Option for Revision Surgery

Christoph Hirche [1,2], Ulrich Kneser [1] and Sebastian Fischer [1,*]

1   Department of Hand, Plastic and Reconstructive Surgery, Microsurgery, Burn Center, BG Trauma Center Ludwigshafen/Rhine, University Heidelberg, Ludwig-Guttmann Str. 13 D, 67071 Ludwigshafen, Germany; christoph.hirche@bgu-frankfurt.de (C.H.); ulrich.kneser@bgu-ludwigshafen.de (U.K.)
2   Department of Hand, Plastic and Reconstructive Microsurgery, BG Unfallklinik Frankfurt am Main gGmbH, Affiliated Hospital to Goethe-University Frankfurt, Friedberger Landstraße 430, 60389 Frankfurt am Main, Germany
*   Correspondence: sfischer@hotmail.de

**Abstract:** Free flaps from the lower abdomen represent the workhorses of modern autologous, microvascular breast reconstruction. Rare signs of venous congestion often become evident during the primary index operation, but a secondary shift of the initially dominant drainage of the DIEV system to the SIEV system with consequent malperfusion is a feared, rare event requiring urgent decision, and complex skill using vein grafts and additional anastomosis to restore a sufficient venous outflow. For secondary anastomosis of the SIEV, especially in stacked flaps, a vein graft to the DIEV or retrograde IMV may be necessary, but this requires an additional donor site, thus prolonging procedure time during the emergency operation and resulting in additional scars of the graft's donor site. We report on a versatile, easy technique of scheduled inframammary fold banking of the dissected, flushed, and clipped non-dominant superficial epigastric vein (SIEV) in unilateral autologous breast reconstruction during the index operation. The banked graft may service as an easy and convenient accessible vein graft in the rare event of secondary shifting of the initial dominant drainage of the DIEV to the SIEV system with the need for urgent re-operation. We retrospectively evaluated the management and outcome of all suitable patients receiving autologous breast reconstruction with a unilateral abdominal DIEP or MS-Tram flap accompanied by banking of the SIEV in the breast pocket between 2017 and 2020 in the present study. In two out of 42 patients (4.8%) receiving autologous breast reconstruction with an abdominal DIEP or MS-TRAM flap with banking of the SIEV in the breast pocket, secondary malperfusion of the flap with progressive venous congestion occurred during the first 48 h perioperatively, between 2 and 37 (mean: 19.5) hours after skin closure. In both cases malperfusion was due to secondary SIEV system dominance, and the banked vein was used as an interpositional graft to the retrograde IMV (case 1) or the DIEV (case 2). Revision surgery lasted between 95 and 121 (mean: 108) minutes without the need for further vein graft harvesting, and further healing was uneventful. Based on the limited cases, inframammary fold banking of the non-dominant SIEV is a versatile, beneficial, and feasible concept with scarce additional dissection time and can be done in all unilateral breast reconstructions to have a reliable graft for emergency re-exploration. It is a useful approach in the context of spare part surgery and tissue banking to safeguard against the rare instance of venous congestion and need for an interpositional graft.

**Keywords:** superficial epigastric vein (SIEV); autologous breast reconstruction; backup option; vein graft; malperfusion; deep inferior epigastric perforator flap (DIEP); congestion

## 1. Introduction

Autologous microvascular breast reconstruction has evolved into a reliable and popular concept with high success rates and quality of life, which has been shown to provide the most natural and long-lasting results [1,2]. The zonal perfusion patterns of the lower abdominal flap have been classified by Hartrampf and Holm and further developed by Saint-Cyr et al. by addressing the clinically relevant perforasomes for surgical dissection and flow management [3–6]. Beyond the increasing evidence on the lower abdomen's perforator flap anatomy, both the unipedicled and bipedicled DIEP or MS-TRAM flap including stacked and/or conjoined flaps with SIEA, SCIA, or DCIA still bear the rare risk of vascular perfusion abnormalities, which may occur directly after dissection or delayed until after restoration of circulation within days [7]. While kinking of the flap pedicle and venous thrombosis can be easily corrected by pedicle repositioning and anastomotic revision with primary venous anastomosis, respectively, a more decisive approach is demanded in the setting of persistent superficial dominance of the venous system via the SIEV. Emergency revisions of the flap with exploration of the pedicle for kinking or thrombosis are required in up to 5.9% of cases, and microsurgical revision is reported to be successful in more than 60% of all cases for autologous breast reconstruction [8,9]. A persistent dominance of the SIEV system was reported to occur rarely in about 0.9% of all cases [10]. In order to address venous insufficiency primarily during the index operation, the superficial venous system of the abdominal flap can be drained via the SIEV by intraflap anastomosis to one of the two accompanying veins of the DIEA, the DIEV, by use as a venous pedicle, as an interpositional graft [11], or by direct anastomosis to a second IMV or the corresponding IMV.

However, a secondary shift of the initially dominant drainage of the DIEV to the SIEV system with venous congestion of the flap and a bloated SIEV occurs in rare cases after the index operation and requires advanced decision making and management. In the case of secondary malperfusion, e.g., with a shortly dissected SIEV, or in stacked, divided, coned, or folded flaps after inset [7,12], a rearrangement of the whole flap anastomosis anatomy can be technically difficult or dangerous without sacrificing the whole flap. Establishing an alternative outflow to the thoracodorsal vessels via direct anastomosis or by use of the cephalic vein as an outflow graft [13] has been described as a reliable backup option to salvage the flap, but results in further scarring and donor site morbidity.

The available donor sites for vein grafts in this particular secondary setting are the cephalic vein as a graft or a venous pedicle, the great saphenous vein, or one of the two DIEV as a graft, while the last one may be limited by distance.

Addressing the concepts of spare part surgery and tissue banking in breast reconstruction [14,15], we report on a feasible and versatile technical step during abdomen in unilateral autologous breast reconstruction by inframammary fold banking of the non-dominant SIEV as an interpositional graft of up to 8 cm to be used in case of secondary congestion during revision due to SIEV drainage dominance after the index operation.

## 2. Materials and Methods

The present study includes a modified approach to stratify the potential risk of secondary venous thrombosis in unilateral microsurgical breast reconstruction with the DIEP or MS-TRAM flap. The management and outcomes of patients treated with inframammary fold banking of the non-dominant superficial epigastric vein (SIEV) in unilateral autologous breast reconstruction were retrospectively evaluated and reported with focus on the perioperative follow-up. The underlying epidemiologic study of the results was in accordance with the declaration of Helsinki and the regulations of the federal ethics committee of Rheinland-Pfalz, Mainz, Germany, without the need for further ethical consultation. Accordance with the local ethical regulations included the presentation of the underlying cases, e.g., anonymized patient demographics, and procedural and outcome-specific parameters without further patient´s consent. The recognizability of patients was excluded.

*Modified approach and technique inframammary fold banking of the non-dominant superficial epigastric vein (SIEV) in unilateral autologous breast reconstruction.*

General harvest of the DIEP or MS-TRAM flap was performed as reported before. As a modification, during dissection of the abdominal flap, bilateral SIEVs were carefully dissected with a length of 6–8 cm and clipped distally. After finalizing the flap design and defining the primary hemiabdominal flap, contralateral perforators were clamped, and both flap perfusion with regard to potential venous congestion and grade of filling of the SIEV are assessed. The contralateral SIEV (Figure 1a) was separated from the remaining hemiflap intended to be disposed, flushed with sterile Heparin solution, and clipped (Figure 1b). Before the flap inset to the breast pocket was finalized, the SIEV graft was placed in the lower part of the breast pocket into the inframammary fold where it was banked as a backup option (Figure 1c,d). In case of revision, the banked SIEV graft could be easily approached in the lower breast pocket and used for intra-flap anastomosis or an alternative recipient site. In bipedicled flaps, both SIEV should be explored, and the non-dominant SIEV of the secondary flap could also be applied as vein graft, albeit this technical innovation is predominantly addressing unipedicled flaps in which a hemiflap or parts of it including its dissected SIEV is intended to be displaced. Dissecting the SIEV up to 6–8 cm should be regarded as an essential step during harvest of the abdominal flap (Figure 1a; Supplementary Video S1), and clipping, flushing, and inframammary fold banking may add a negligible 2 min to the whole procedure (Figure 1b–d; Supplementary Video S1).

(**a**)

(**b**)

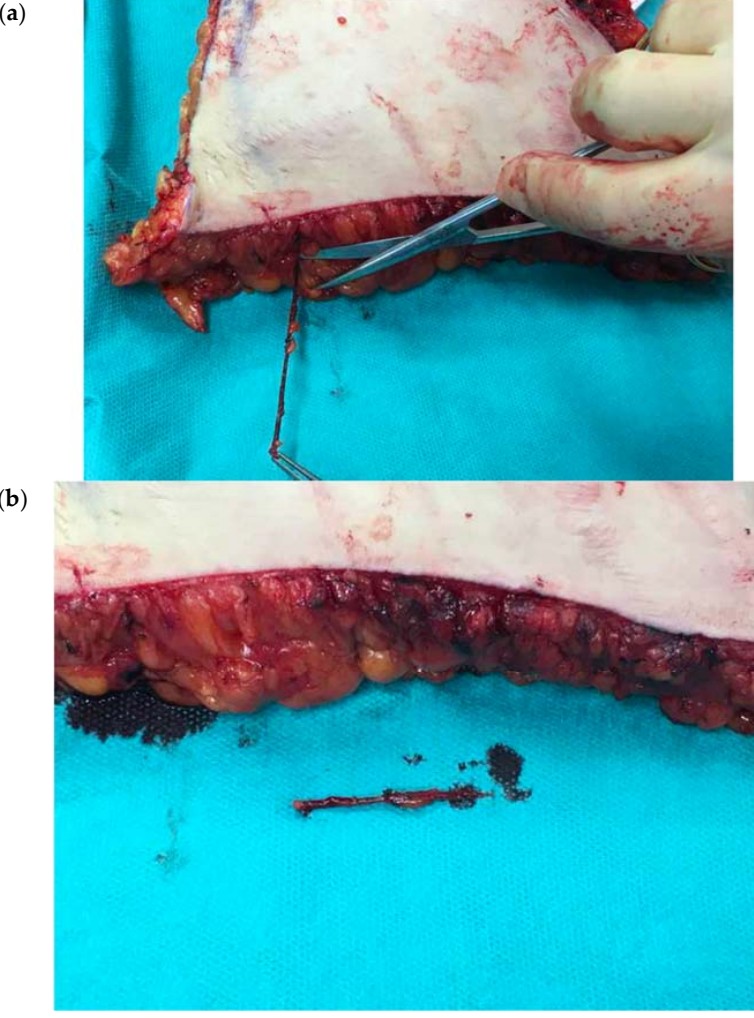

**Figure 1.** *Cont.*

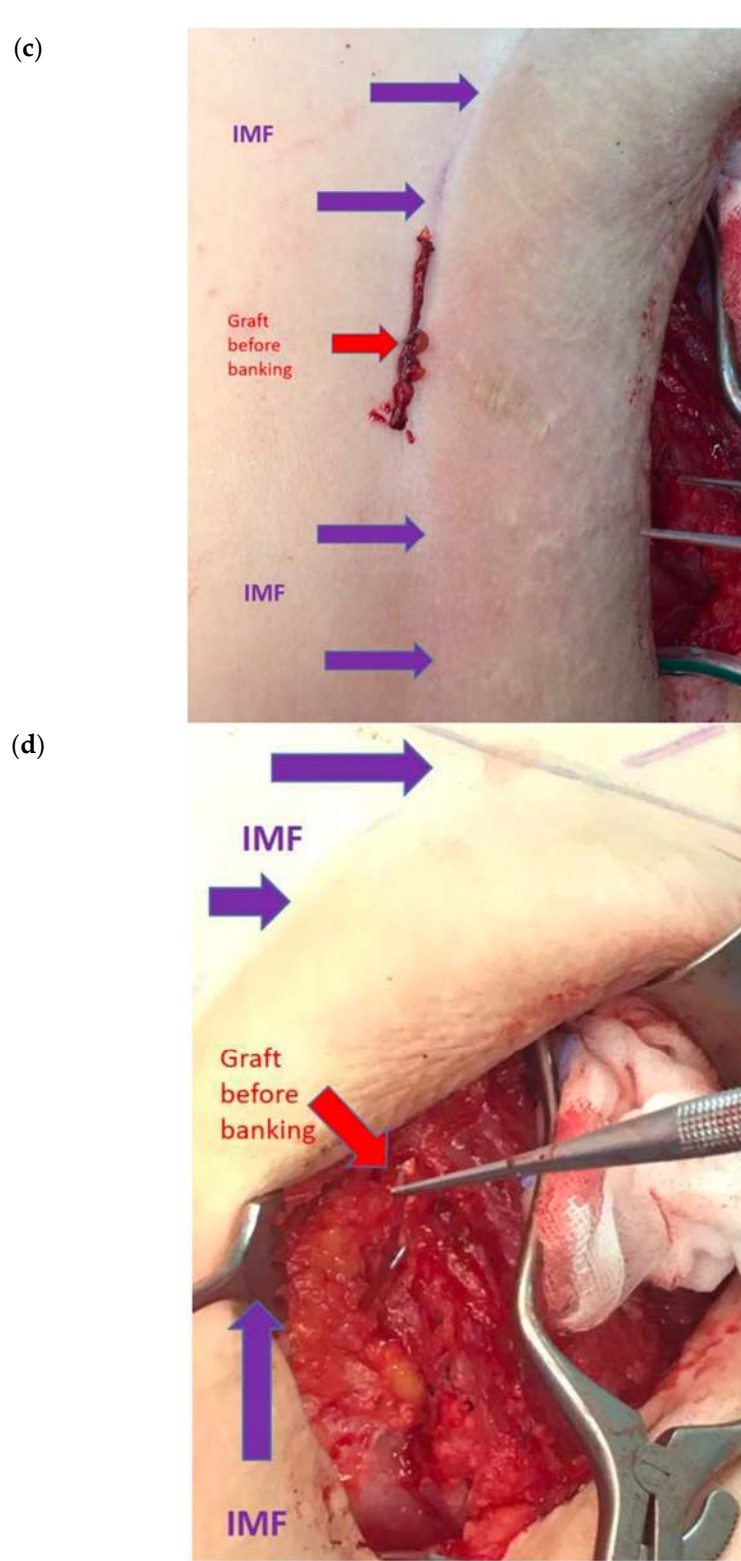

**Figure 1.** The technical steps of using the non-dominant superficial epigastric vein (SIEV) as a graft of up to 8 cm for inframammary fold banking are displayed. In case of finalizing the hemiabdominal flap, the contralateral SIEV (**a**) was separated from the remaining flap and spared, and then flushed with sterile heparin solution and clipped (**b**). Before finalizing the inset of the flap to the breast pocket, the SIEV graft was placed in the lower part of the breast pocket close to the inframammary fold where it is banked as a backup option (**c**,**d**).

## 3. Results

In 42 eligible patients with unilateral microsurgical breast reconstruction using the abdominal DIEP or MS-TRAM flap between 2017 and 2020, inframammary fold banking of the non-dominant superficial epigastric vein (SIEV) was done during the index operation. All breast reconstructions were successfully finalized by flap positioning in the remaining breast envelope with subsequent shaping of the breast in relation to the contralateral breast, pedicle positioning, and final closure of the skin by resorbable sutures.

In two out of 42 patients, (4.8%) secondary malperfusion of the flap with progressive venous congestion occurred during the first 48 h perioperatively. Clinical detection of the malperfusion by capillary refill test and overall flap assessment occurred between 2 and 37 (mean: 19.5) hours after skin closure.

Emergency re-exploration of each flap was done, and in both cases the exploration of the arterial anastomosis and the two venous anastomoses did not show thrombotic occlusion or kinking. Secondary exploration of the dominant SIEV vessel primarily included in the flap revealed significant dilatation and congestion, and reestablishing the drainage by removing the microsurgical clips showed acute relief of venous congestion. Anticipating a secondary shift of the dominance of the venous drainage from the DIEV to the SIEV system, the 6–8 cm dominant SIEV was prepared for reconstruction: the banked non-dominant SIEV graft was taken from the inframammary, re-opened and flushed for patency testing. After approval, it was used as an interpositional vein graft between the ipsilateral dominant SIEV for distal anastomosis and the retrograde IMV (case 1) and the DIEV to re-establish greater length (case 2) with each two coupler anastomoses.

Revision surgery lasted between 95 and 121 (mean: 108) minutes. In both cases the graft allowed optimal handling of the anastomosis and flap arrangement for breast shaping, and flap perfusion was restored in both cases without further compromise of the flap. In both cases, unilateral breast reconstruction was successful, and further donor sites with scarring as well as additional operation time for harvesting the graft in the emergency situation were unnecessary due to the banked non-dominant SIEV vessel in the inframammary.

## 4. Discussion

Although current approaches and algorithms to perforator selection for the DIEP flap and the optional use of MS-TRAM flaps have allowed microsurgeons to consistently select appropriate perforators and constant recipient vessels—making venous congestion a rare occurrence—venous malperfusion can lead to reconstructive failure in breast reconstruction. Unexpected perfusion patterns of the abdominal flap, especially in relation to the superficial and deep fat tissue layer and its corresponding vasculature, are a rare but challenging situation. Delayed venous congestion due to secondary dominance of the superficial system in particular requires versatile and reliable backup options. In the case of secondary malperfusion [7,12], a rearrangement of the whole flap anastomoses anatomy and local search for additional recipient veins, as the corresponding IMV or an IMV perforator, can be technically challenging with size mismatch, or impossible, leading to flap loss. Establishing an alternative outflow to the thoracodorsal vessels via direct anastomosis, or use of the cephalic vein as an outflow graft [13] have been suggested to salvage the flap with a safe venous axis, but results with further scarring and donor site morbidity have to be taken into consideration. The presented technical refinement of routine inframammary fold banking of the non-dominant SIEV in the inferior pocket during unilateral autologous breast reconstruction is a simple and helpful backup option in the context of spare part surgery and tissue banking and eliminates the need for a secondary donor site. An alternative vein that could serve the same purpose and concept and would be also available in bilateral reconstructions is the SCIV.

The banked vein can be taken out of the pocket, re-flushed, and then used as an interpositional vein graft, e.g., to be anastomosed between one of two accompanying DIEV of the flap to the SIEV. To our knowledge, it does not contradict any tissue bank

regulation worldwide, which is a remarkable advantage. Based on the tissue diffusion in the inframammary pocket, usage for revision in the first 72 h may be feasible. Compared to an emergency harvest of the cephalic vein or great saphenous vein during revision, the additional work and time is not relevant, and a secondary donor site can be avoided. Limitations may include size mismatch, which rarely occurs, and inappropriate length, e.g., in complex revision, when an additional graft has to be harvested in any case.

Follow-up of patients in this study was focused on the perioperative period for flap perfusion and survival, which can certainly be assessed in the given time frame. Nevertheless, the follow-up of patients is too short to analyze the overall outcome of breast reconstruction in this cohort, which was not the aim of this study. Future studies should provide a longer follow-up of patients and focus on patient satisfaction, aesthetic, and quality of long-term life outcomes using IMF baking.

With forty-two patients and two relevant events of patients who required revision surgery for venous congestion and a vein graft, the number of cases presented in this study is limited and does not provide solid evidence, as revision for anastomosis in microsurgical breast reconstruction is rather low. However, we believe that the number of patients and events is sufficient to introduce this novel approach as an option for flap takeback and successful salvage and its proof of principle. Nevertheless, more patients are necessary to substantiate our findings.

## 5. Conclusions

Decisive bilateral dissection of both SIEV even for unilateral breast reconstruction up to a length of 8 cm enables intraoperative evaluation of the superficial venous drainage dominance of the abdominal flap. Inframammary fold banking of the non-dominant SIEV instead of discarding can be done in all unilateral cases for breast reconstruction to have a reliable graft for any anastomosis to salvage the flap in DIEP or MS-TRAM flap based unilateral autologous breast reconstruction. It is a useful approach to safeguard against the rare instance in which one is in need of a vein graft to expeditiously correct venous congestion. This surgical refinement is a simple technique that in select cases tremendously simplifies surgical treatment of a rare complication with negligible impact on surgical duration.

**Supplementary Materials:** The following supporting information can be downloaded at: https://www.mdpi.com/article/10.3390/std11010004/s1, Supplementary Video S1: The sequence demonstrates full-length harvest of the SIEV and banking in the inframammary fold.

**Author Contributions:** Conceptualization, C.H. and U.K.; methodology, C.H.; validation, C.H., U.K. and S.F.; formal analysis, C.H.; writing—original draft preparation, C.H.; writing—review and editing, S.F.; All authors have read and agreed to the published version of the manuscript.

**Funding:** This research received no external funding.

**Institutional Review Board Statement:** The presented case study is in accordance with regulations of the ethics committee of the state chamber of Medicine in Rheinland-Pfalz, Mainz, Germany and is in accordance with the ethical standards as laid down in the 1964 Declaration of Helsinki and its later amendments or comparable ethical standards.

**Informed Consent Statement:** Not applicable.

**Data Availability Statement:** We thereby declare that we have provided all raw data on which your study is based in the present paper.

**Conflicts of Interest:** The authors declare no conflict of interest.

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
