# Peer review of "Inframammary Fold Banking of the Non-Dominant Superficial Epigastric Vein (SIEV) in Unilateral Autologous Breast Reconstruction: A Simple and Helpful Backup Option for Revision Surgery"

_2038-9582, doi:10.3390/std11010004_

Round 1

Reviewer 1 Report

The authors presented a new technical refinement for microsurgical breast reconstruction in dealing with venous congestion after operation. Banking of vein graft reduces the operation time and venous graft harvesting donor site scar. However, the case number is too small for providing solid evidences  of the viability & patency of the banked vein graft and the successfulness of revision  by this technique.  

As the title of the manuscript declared, the authors provide a simple and helpful backup option for revision surgery in unilateral autologous breast reconstruction. The only problem is the case number of this report. However, this is a nice and helpful idea for decrease the donor site morbidity/ scar when vein graft is needed in revision surgery. 

Author Response

Reviewer 1

The authors presented a new technical refinement for microsurgical breast reconstruction in dealing with venous congestion after operation. Banking of vein graft reduces the operation time and venous graft harvesting donor site scar. However, the case number is too small for providing solid evidences of the viability & patency of the banked vein graft and the successfulness of revision  by this technique.

As the title of the manuscript declared, the authors provide a simple and helpful backup option for revision surgery in unilateral autologous breast reconstruction. The only problem is the case number of this report. However, this is a nice and helpful idea for decrease the donor site morbidity/ scar when vein graft is needed in revision surgery. 

We want to thank this reviewer for this supportive comment. We totally agree that the number of cases is rather low. However, for a novel approach for flap salvage back-up and proof of principle 42 patients is a good start. Nevertheless, we included this important aspect in the limitations section of the manuscript as follows:

 “With forty-two patients and two relevant events of patients, who required revision surgery for venous congestion and a vein graft, the number of cases presented in this study is limited and not providing solid evidence, as revision for anastomosis in microsurgical breast reconstruction is rather low. However, we believe that the number of patients and events is sufficient to introduce this novel approach as an option for flap take back and successful salvage and its proof of principle. Nevertheless, more patients are necessary to substantiate our findings.”

Reviewer 2 Report

Inframammary fold banking of the non-dominant superficial epigastric vein (SIEV) in unilateral autologous breast reconstruction: a simple and helpful backup option for revision surgery

DIEP flap breast surgery is now a gold standard in autologous breast reconstruction. Although the reconstruction possibilities are abundant, implant-based, perforator flaps, and other free flaps, in the majority DIEP flap gives us the best results and the donor site morbidity is the less conspicuous.

As we know the major problem in free flap reconstruction is venous congestion, which gives us a problematic decision-making approach. In some cases, the congested flap cannot be salvaged.

The idea of banking spare tissue was used over the years, but the idea of banking the additional vein graft for a salvage operation in the first 72 hours is reasonable and tempting.

The authors describe 42 patients and only 2 cases of flap congestion which is a great result. Although in those two cases the flaps could be salvaged by using the banked vein.

I find this study interesting well organized, well written, and well prepared. The described patient groups are well defined. Despite the low volume of the patents in which the procedure was performed, the study presents a good and versatile option of flap salvage.

On the other hand, we cannot compare and predict how repetitive it is due to small volume of revision cases.

I have three pleas:

  1. The lack of statistical power
  2. The lack of the follow-up
  3. Minor language issues

Due to a presentation of the technical novelty, I recommend this manuscript be published as an original article.

Author Response

Reviewer 2

Inframammary fold banking of the non-dominant superficial epigastric vein (SIEV) in unilateral autologous breast reconstruction: a simple and helpful backup option for revision surgery

DIEP flap breast surgery is now a gold standard in autologous breast reconstruction. Although the reconstruction possibilities are abundant, implant-based, perforator flaps, and other free flaps, in the majority DIEP flap gives us the best results and the donor site morbidity is the less conspicuous.

As we know the major problem in free flap reconstruction is venous congestion, which gives us a problematic decision-making approach. In some cases, the congested flap cannot be salvaged.

The idea of banking spare tissue was used over the years, but the idea of banking the additional vein graft for a salvage operation in the first 72 hours is reasonable and tempting.

The authors describe 42 patients and only 2 cases of flap congestion which is a great result. Although in those two cases the flaps could be salvaged by using the banked vein.

I find this study interesting well organized, well written, and well prepared. The described patient groups are well defined. Despite the low volume of the patents in which the procedure was performed, the study presents a good and versatile option of flap salvage.

On the other hand, we cannot compare and predict how repetitive it is due to small volume of revision cases.

I have three pleas:

  1. The lack of statistical power
  2. The lack of the follow-up
  3. Minor language issues

Due to a presentation of the technical novelty, I recommend this manuscript be published as an original article.

  1. We want to thank this reviewer for this supportive comment. We totally agree that the number of cases is rather low. However, for a novel approach for flap salvage back-up and proof of principle 42 patients is a good start. Nevertheless, we included this important aspect in the limitations section of the manuscript as follows:

 “With forty-two patients and two relevant events of patients, who required revision surgery for venous congestion and a vein graft, the number of cases presented in this study is limited and not providing solid evidence, as revision for anastomosis in microsurgical breast reconstruction is rather low. However, we believe that the number of patients and events is sufficient to introduce this novel approach as an option for flap take back and successful salvage and its proof of principle. Nevertheless, more patients are necessary to substantiate our findings.” 

  1. Indeed, follow-up of patients is rather short. However, primary outcome was flap survival, which can certainly be assessed in the given time frame. However we included this important aspect in the limitations section of the manuscript as follows:

“Follow-up of patients in this study was focused on the perioperative period for flap perfusion and survival, which can certainly be assessed in the given time frame. Nevertheless, follow- up of patients is too short to analyze the overall outcome of breast reconstruction in this cohort, which was not the aim of this study. Future studies should provide a longer follow-up of patients and focus on patient satisfaction and aesthetic and quality of life long-term outcomes using IMF banking.” 

  1. We totally agree with this reviewer and apologize for minor language issues. A native speaking physician proofread the manuscript and we revised it accordingly.

English language revisions are made throughout the whole manuscript.

Reviewer 3 Report

In case of venous congestion requiring urgent interpositional graft in unilateral autologues breast reconstruction revision surgery, using the prepared inframammary fold banking of the non-dominant superficial epigastric vein (SIEV) is a good idea. It is a simple and helpful backup option for this kind of emergency situation.  

Author Response

Reviewer 3

In case of venous congestion requiring urgent interpositional graft in unilateral autologues breast reconstruction revision surgery, using the prepared inframammary fold banking of the non-dominant superficial epigastric vein (SIEV) is a good idea. It is a simple and helpful backup option for this kind of emergency situation.

We want to thank this reviewer for this supportive comment.
